# The Role of High-Dose-Rate Brachytherapy (Interventional Radiotherapy) in the Reirradiation of Liver Metastases

**DOI:** 10.3390/cancers17244013

**Published:** 2025-12-16

**Authors:** Paweł Cisek, Izabela Kordzińska-Cisek, Aleksandra Kozłowska, Ludmiła Grzybowska-Szatkowska

**Affiliations:** 1Department of Radiotherapy, Medical University of Lublin, 20-081 Lublin, Poland; akozlowska@usk1.pl (A.K.); ludgr@poczta.onet.pl (L.G.-S.); 2Department of Radiotherapy, Oncology Center of Lublin, 20-090 Lublin, Poland; ikordzinska@cozl.pl

**Keywords:** reirradiation, brachytherapy HDR, interventional radiotherapy, liver metastases, toxicity

## Abstract

This study evaluated outcomes, prognostic factors and toxicity of high-dose-rate (HDR) brachytherapy reirradiation for liver metastases in 59 oligometastatic patients previously treated with stereotactic body radiation therapy (SBRT) or HDR brachytherapy. Local control (LC), progression-free survival (PFS) and overall survival (OS) were analysed in relation to clinical factors, treatment indication, tumour characteristics, dose and response. Hepatic toxicity was assessed based on dose, treated volume, number of lesions, interval since prior radiotherapy and dose to uninvolved liver. After a median 13-month follow-up, median LC, PFS and OS were 9, 8 and 13 months. Partial regression, stable disease and progression occurred in 32%, 44% and 12% of patients. Tumour shrinkage most strongly influenced LC, PFS and OS; fewer metastases and limited prior systemic therapy also improved outcomes. Toxicity was low, with no strong link between dosimetric and biochemical liver parameters.

## 1. Introduction

The liver is one of the most common sites for metastases from solid tumours, accounting for almost a quarter of all cancers [1]. It is also responsible for over 90% of cancer-related deaths [2]. While all cancers can metastasise to the liver, colorectal, breast, prostate and pancreatic cancers are the most commonly affected [2]. In the case of colorectal cancer, the liver is the main site of metastasis. Metachronous metastases are common, often resulting from inadequate initial or adjuvant treatment [3]. The mainstay of treatment for metastatic disease is systemic therapy. However, in the case of oligometastatic disease—defined in most publications as a maximum of five metastases—local methods are playing an increasingly important role [4,5]. Oligometastatic disease is not uniform; the ESTRO classification distinguishes nine clinical scenarios. These differ in terms of the time of onset, the initial number of metastases, and the response to previous treatment [4]. Among them, we distinguish repeat oligoprogression (a condition in which oligoprogression occurs in oligometastatic disease with no more than five metastases), induced oligoprogression (a condition in which oligoprogression occurs in polymetastatic disease with no more than five metastases) and oligopersistence disease (a condition in which active metastatic sites occur in no more than five metastases despite regression after systemic treatment) [4]. The aim of local treatment for oligometastatic disease is to extend the time until the next systemic treatment is required, prolong survival rates and potentially cure patients [5]. The basis of local treatment for liver metastases is their surgical removal. There are also many non-surgical methods of local treatment for liver metastases, such as thermal ablations (e.g., radiofrequency ablation (RFA) and laser interstitial thermal therapy (LITT)), chemoembolisation (transarterial chemoembolisation (TACE)), radioembolisation (yttrium-90 radioembolisation (Y-90 RE)), and stereotactic radiotherapy (SBRT) [6]. However, many of these techniques are limited by the size, number and location of metastases, as well as by the presence of anatomical structures that pose a barrier to their safe use. A commonly used radiotherapy technique is stereotactic body radiation radiotherapy (SBRT) [7]. It is non-invasive, but its application is limited by the doses administered to critical organs. This is particularly important in cases involving large or multiple metastases. High dose rate (HDR) brachytherapy is a more invasive method that is used less frequently. Thanks to the rapid decrease in dose with distance from the radiation source, it is possible to deliver a high dose to the tumour while delivering a relatively low dose to adjacent organs [8].

Despite the high local efficacy of both irradiation methods, tumour progression is common. Repeated SBRT is often limited by the dose to critical organs, primarily the liver. In this context, repeated brachytherapy is a valuable alternative. European Society for Radiotherapy and Oncology and European Organisation for Research and Treatment of Cancer (ESTRO/EORTC) recommendations distinguish between two types of reirradiation [9]. In the first type, the irradiation areas overlap geometrically. In the second type, they do not overlap; however, due to their proximity, there is a real concern for increased toxicity in the irradiated organ.

Studies examining the role of SBRT in reirradiation primarily focus on hepatocellular carcinoma. There is limited data on the reirradiation of liver metastases. In a study by McDuff et al. [10], the median overall survival was 14 months, and the treatment failure rate after one year was 46.4%. In a Korean study, partial regression was observed in 63% of patients after a 14-month follow-up period, complete regression in 19%, and the median overall survival was 11 months [11]. In the Huang et al. study [12], SBRT reirradiation was performed after a median of one month without disease progression. Although over one-third of patients developed radiation-induced liver disease (RILD), the median overall survival in patients without RILD was 29 months.

However, there are no data on the use of repeated brachytherapy after prior SBRT or brachytherapy. This study aimed to retrospectively analyse the preliminary results of repeated HDR brachytherapy treatment for liver metastases from various cancers in the oligometastatic stage. Another aim was to determine the prognostic factors and toxicity of brachytherapy treatment.

## 2. Materials and Methods

### 2.1. Patient Selection Criteria

The study is part of a retrospective single-centre analysis of 574 patients undergoing brachytherapy for primary or secondary liver cancers, treated at the Department of the Lublin Region Oncology Centre between 2015 and 2024. The study included 59 patients with liver metastases from various solid tumours, who underwent radiotherapy using HDR brachytherapy or SBRT, followed by reirradiation to the same area (type 1 reirradiation) or to a different area but within the same organ—liver (type 2 reirradiation) [9]. Treatment was indicated for oligometastatic disease, as defined by Guckenberg et al. [4]. Reirradiation with brachytherapy was performed in patients with repeat oligoprogression or induced oligopersistence (type 2 reirradiation only) or induced oligoprogression (type 1 or 2 reirradiation). Inclusion criteria for the study included a World Health Organization (WHO) performance status score of less than or equal to 2; haematological stability (haemoglobin (HGB) greater than or equal to 8 mg/dL, white blood cell (WBC) count greater than or equal to 2000/mm^3^, neutrophil (NEU) count greater than or equal to 1500/mm^3^, platelet (PLT) count greater than or equal to 50,000/mm^3^); healthy liver and kidney function (alanine aminotransferase (ALT), aspartate aminotransferase (AST) and total bilirubin (BIL) levels less than or equal to 2.5 times the upper limit of normal; creatinine level less than or equal to 2 mg/dL); a maximum diameter of metastases up to 15 cm; and a maximum of five metastases. Patients were excluded if metastases were resectable, if there was an inability or contraindication to liver applicator placement, or if metastases were located <1 cm from the wall of the stomach, small intestine, or colon. All patients received systemic treatment according to the applicable standards for the given tumour. Systemic treatment within the same line of therapy was continued in patients with repeat or induced oligoprogression. Patients with induced oligopersistence did not receive continued systemic treatment and were observed instead. Disease progression was confirmed using at least two imaging modalities: CT with MRI or PET-CT. For type 1 reirradiation, the minimum interval from the first radiation treatment had to be at least 9 months.

### 2.2. Cohort Characteristics and Treatment

The patients were analysed in terms of the following factors:Epidemiological: sex and age.Clinical: type of cancer, general condition according to the WHO classification, location, number and size of metastases, line of systemic treatment, type of previous radiotherapy, time from radiotherapy to tumour progression, time to re-irradiation and indication for irradiation.Dosimetric: total dose, dose in two-thirds of the non-irradiated liver volume (D2/3) and dose in 700 cm^3^ of the non-irradiated liver volume (D700 cm^3^).

Due to the variety of irradiation schedules prior to re-irradiation, all doses were converted to Biological Effective Dose (BED) according to the assumptions of the linear-quadratic model [13]. Due to the variety of tumours, mainly glandular, the alpha-beta coefficient was assumed to be 5 for the tumour and 3 for the liver. Data were collected on biochemical liver function parameters, such as total bilirubin, alanine aminotransferase, aspartate aminotransferase and albumin. Values were determined for all parameters before brachytherapy, at the maximum value within 12 weeks after brachytherapy, and then at three-monthly intervals (except for albumin, for which the minimum value was determined within 12 weeks). Possible clinical features and biochemical liver damage were also analysed in all patients according to the Child-Pugh classification [14]. Based on the albumin and bilirubin levels, the ALBI index was calculated [15]. Analysis of the biochemical parameters was performed using the immunoenzymatic method with the Beckman Coulter AU5800 apparatus (Brea, CA, USA).

The application was performed under the guidance of continuous tomographic imaging using a Somatom Confidence tomograph (Siemens, Berlin, Germany). Doses of 15, 20 and 25 Gy were administered. The dose was specified at the 95% isodose level. The dose level depended on the dose to critical organs. The following criteria for critical organ protection were applied: The most important critical organ was the liver (D tolerance = D2/3 < 5 Gy), followed by the stomach (D tolerance = D1cm3 < 15 Gy), the gallbladder (D tolerance = D1max < 20 Gy), the intestines (D tolerance = D1cm3 < 12 Gy) and the kidney (V7Gy < 2/3 of the total volume). Patient characteristics are presented in Table 1. As the primary dose limitation was D2/3 <5 Gy, this parameter was converted to BED with an alpha/beta ratio of 3 for single-fraction administration (13.33 Gy). The doses from the individual irradiation cycles (the first cycle and the re-irradiation cycle) were converted to BED and then added together (see Table 2) The treatment plan was accepted on the condition that the BED did not exceed 13.33 Gy for any of the treatment cycles). Additionally, the dose was determined in 700 cm^3^ of non-irradiated liver tissue (D700 cm^3^), administered in three fractions (BED < 40 Gy). Due to the low doses received by other critical organs, which did not exceed tolerance levels, conversion of doses for other organs at risk (OARs) was not necessary. Treatment planning was performed using Brachyvision ver. 13 treatment planning system (Varian, Palo Alto, CA, USA).

### 2.3. Follow Up

During the post-treatment period, patients underwent regular imaging studies, including computed tomography (CT) or magnetic resonance imaging (MRI) scans (four to six times per year). The Response Evaluation Criteria in Solid Tumours (RECIST) 1.0 criteria were used to evaluate the response to treatment. Due to the risk of pseudoprogression in the early post-radiation period (up to 6 weeks), CT scans were used to assess treatment response at least 12 weeks after brachytherapy (following two consecutive scans performed at least 4 weeks apart) and no later than 5 months afterwards. An abdominal MRI scan was also performed to confirm or exclude pseudoprogression. Patients were assigned to the appropriate response category according to RECIST criteria based on changes in metastasis size on CT scans. If an MRI scan was performed, the assessment of size changes and functional sequences (Diffusion Weighted Imaging—DWI) were both used to evaluate the response. Toxicity was assessed using the Common Terminology Criteria for Adverse Events (CTCAE) scale, version 5.0.

### 2.4. Statistical Analysis

A survival analysis was performed using the Kaplan–Meier method. Overall survival (OS) was defined as the time from reirradiation to patient death or the end of the follow-up period. Progression-free survival (PFS) was defined as the time from reirradiation to the progression of any metastasis (irradiated or not). Local control (LC) was defined as the time from reirradiation to progression of an irradiated metastasis. The Log-rank test was used to analyse the factors influencing LC, PFS and OS. Cox proportional regression analysis was used to analyse the influence of prognostic factors on LC, PFS and OS. To examine the relationships between variables, the non-parametric U Mann–Whitney test was used for independent variables to compare differences between two patient groups, and the Kruskal–Wallis test was used to compare differences between multiple patient groups. The Pearson correlation test was used to determine the relationships between variables. A positive correlation value meant that as one variable increased, so did the other. Conversely, a negative correlation value meant that as one variable increased, the other decreased. Values below 0.2 or above −0.2 were considered to indicate no correlation; values in the range −0.2 to −0.4 or 0.2 to 0.4 were considered to indicate a weak correlation; values in the range −0.4 to −0.7 or 0.4 to 0.7 were considered to indicate a moderate correlation; values in the range −0.7 to −0.9 or 0.7 to 0.9 were considered to indicate a fairly strong correlation; and values below −0.9 or above 0.9 were considered to indicate a very strong correlation. *p* < 0.05 was also considered to indicate a statistically significant difference. Statistical analysis was performed using Statistica version 13 software (Statsoft, Tulsa, OK, USA).

All procedures performed in studies involving human participants were in accordance with the ethical standards of the institutional research committee and with the 1964 Helsinki Declaration and its later amendments or comparable ethical standards. The study was approved by the Lublin Medical Chamber no. LIL-KB-20/2014. Written consent was obtained from each patient.

## 3. Results

### 3.1. Analysis of Survival Parameters and Degree of Response to Treatment

The median follow-up was 13 (range 2–52) months. The 6-month overall survival (6m-OS) rate was 93%, and the median overall survival (mOS) was 14 months. The 6-month local control (6m-LC) rates for patients initially irradiated, undergoing their first reirradiation, and their second reirradiation were 100%, 80% and 97%, respectively. Median local control (mLC) after the first and second reirradiation was 13 and 9 months. These outcomes differed significantly between reirradiated patients and those treated initially with radiotherapy (mLC in the latter group was 16.3 months; Log-rank 9.93, *p* = 0.007). The 6-month progression-free survival (6m-PFS) was 100%, 73% and 73% in patients initially irradiated, after the first reirradiation, and after the second reirradiation, respectively. Median PFS (mPFS) was 10.3 months for primary radiotherapy, 8 months for first reirradiation, and 9 months for second reirradiation. These differences were not statistically significant (Log-rank 1.08, *p* = 0.583). Figure 1 shows the Kaplan–Meier curves for OS in all patients and for PFS and LC in patients who received subsequent cycles of irradiation.

In the analysed group, the disease control rate was 88%, with disease progression occurring in seven patients (progression disease (PD)—12%). The objective response rate was 32% (19 patients), all of whom achieved a partial response (PR). The remaining 26 patients (44%) achieved stable disease (SD). The response rate was significantly dose-dependent. The greater the dose, the greater the response (Kruskal–Wallis test value = 9.563, *p* = 0.008). Figure 2 shows the dose-dependent response to treatment. The type of reirradiation did not affect the depth of response (Mann–Whitney U-test value = 1.454, *p* = 0.228). There was also a moderate correlation between the percentage of tumour shrinkage and OS. The greater the tumour regression, the longer the OS (Pearson test value = −0.469, *p* < 0.001).

### 3.2. Risk Factor Analysis

The influence of various factors on LC, PFS and OS was analysed. In univariate analysis, clinical diagnosis, line of systemic treatment, number of metastases, dose and percentage tumour shrinkage were found to have a significant impact on LC. However, in multivariate analysis, only the percentage tumour shrinkage was found to be an independent factor influencing LC (Table 3 and Table 4).

#### 3.2.1. Local Control

Patients with colorectal cancer had poorer local control than those with other cancers. The mLC for patients with colorectal cancer was 9 months; for breast cancer patients, it was 14 months; and for patients with other cancers, it was 18 months (Log-rank test value = 6.61, *p* = 0.037). Best LC was seen with brachytherapy in the first two lines of treatment. Subsequent lines of treatment showed an mLC of 17, 24, 9, 7 and 3 months (Log-rank test value = 10.036, *p* = 0.039). There was also a statistically significant difference in the number of treated metastases: 14 and 6 months for one metastasis and more than one metastasis, respectively (Log-rank test value = 2.150, *p* = 0.032). Dose also impacted LC: 15 months for 25 Gy, 10 months for 20 Gy and 7 months for 15 Gy (Log-rank test value = 8.034, *p* = 0.18). The longest LC was found in patients who achieved partial response at 12 months. In patients with stable disease and progressive disease, the mLC was 7 and 3 months (Log-rank test value = 10.201, *p* = 0.006). Figure 3 shows a statistically significant correlation between LC and the type of cancer (Figure 3A), lines of systemic treatment (Figure 3B), number of metastases (Figure 3C), total dose (Figure 3D) and response to treatment (Figure 3E).

#### 3.2.2. Progression-Free Survival

Progression-free survival depended on the WHO performance status. Patients with a good performance status had a longer PFS. The mPFS in WHO stages 0, 1 and 2 was not reached, 11 months and 6 months (Log-rank test value = 8.643, *p* = 0.013). Patients with repeat oligoprogression and induced oligopersistence had a better prognosis than those with induced oligoprogression, with mPFS of 11, 9 and 7 months (Log-rank test value = 6.053, *p* = 0.048). Treatment in later lines of therapy also worsened the prognosis. The mPFS in lines 1, 2, 3, 4 and 5 were 17, 12, 7, 7 and 4.5 months (Log-rank test value = 9.661, *p* = 0.046). Patients whose disease was limited to the liver had a better prognosis, with a Log mPFS of 10 months compared to 7 months for those with concomitant liver metastases (Log-rank test value = 2.26, *p* = 0.024). There was a statistically significant difference in the number of treated metastases: mPFS was 11 and 6 months for patients with one or more metastases (Log-rank test value = 2.486, *p* = 0.013). The RECIST response rate also influenced PFS: mPFS for partial response, stable disease, and progressive disease were 14, 8, and 3 months (Log-rank test value = 25.299, *p* < 0.001). Figure 4 shows a statistically significant correlation between PFS and the WHO performance status (Figure 4A), indication of treatment (Figure 4B), line of systemic treatment (Figure 4C), extrahepatic metastases (Figure 4D), number of metastases (Figure 4E) and response to treatment (Figure 4F).

#### 3.2.3. Overall Survival

The effect of performance status on overall survival was statistically significant. Median OS for WHO stages 0, 1, and 2 was 52, 16, and 11 months, respectively (Log-rank test value = 5.80, *p* = 0.49). Brachytherapy treatment in later lines of systemic treatment worsened prognosis (mOS in lines 1, 2, 3, 4 and 5 were 34.39, 13, 11 and 6 months, respectively, Log-rank test value = 23.287, *p* < 0.001). Patients whose disease was limited to the liver had a better prognosis, with an mOS of 16 months compared to 11 months for those with concomitant liver metastases (Log-rank test value = 2.384, *p* = 0.017). A statistically significant deterioration in OS was only observed in the case of lung metastases. The mOS in patients with lung metastases was 11 months and 16 months in patients without lung metastases (Log-rank test value = 2.202, *p* = 0.028). A statistically significant difference was also found in the number of treated metastases. The mOS in patients with one metastasis and in patients with more than one metastasis was 19 and 11 months, respectively (Log-rank test value = 2.359, *p* = 0.018). A high dose was found to have a statistically significant effect on OS: the mOS for doses of 25 Gy, 20 Gy and 15 Gy was 14 months, 7 months and 6 months, respectively (Log-rank test value = 15.565, *p* < 0.001). The degree of response to treatment according to the RECIST scale also influenced PFS: mPFS for partial response, stable disease and progressive disease were 28, 14 and 6 months, respectively (Log-rank test value = 13.455, *p* = 0.001).

Figure 5 shows a statistically significant correlation between OS and WHO performance status (Figure 5A), line of systemic treatment (Figure 5B), extrahepatic and lung metastases (Figure 5C,D), number of metastases (Figure 5E), total dose (Figure 5F) and response to treatment (Figure 5G).

### 3.3. Toxicity Analysis

The rate of surgical complications in the analysed group of patients was low. Asymptomatic bleeding, visible as a small haematoma on the postoperative CT scan, was observed in eight patients (13.5%). Three patients (5%) also experienced haemoglobin depletion of more than 2 g/dL. One of these patients required a transfusion. One patient had clinically insignificant pneumothorax on a postoperative CT scan, which did not require treatment. Fourteen patients (25%) reported pain at the puncture site, requiring analgesics. One patient had radiation gastritis, confirmed by gastroscopy, and two patients had a radiation skin reaction at the applicator puncture site. No clinical signs of liver damage or gastrointestinal reactions were observed. The median baseline total bilirubin level was 0.8 mg/dL (range 0.4–1.7 mg/dL), alanine aminotransferase (ALT)—32 IU/L (range 12–65 IU/L), aspartate aminotransferase (AST)—27 IU/L (range 10–54 IU/L), and albumin level—4 g/dL (range 2.5–5.4 g/dL). Following treatment, the median increase in bilirubin level was 0.3 mg/dL (range 0.1–1.1 mg/dL), ALT increased by 17 IU/L (range 1–133 IU/L), and AST increased by 4.5 IU/L (range 1–48 IU/L). The median difference in albumin level was 0.15 g/dL (−0.8–1.6 g/dL). All patients were in Child-Pugh class A at baseline. No patient experienced a deterioration in liver function to grade B or C following treatment. The median ALBI score was −2.618 (−3.551 to −1.764) before treatment and −2.593 (−3.296 to −1.799) after treatment. Before treatment, 32 patients (54%) had a grade 1 ALBI score, and 27 patients (46%) had a grade 2 ALBI score. Following treatment, eight patients (13.5%) experienced an increase in their ALBI score from grade 1 to grade 2. Detailed toxicity data are presented in Table 5. No acute complications higher than grade 3 were observed.

Table 6 presents the Pearson coefficient value alongside the *p*-value. The correlation of biochemical parameters of liver function after brachytherapy treatment and the differences between the values before and after treatment with dosimetric parameters were tested. The biochemical parameters of liver function were ALT, AST, bilirubin, albumin and ALBI. The dosimetric parameters, however, are metastasis size and volume, total dose, D2/3 and D700 cm^3^ liver volumes, and biologically effective dose for total dose, D2/3 and D700 cm^3^. No strong correlation was found between these parameters. The only statistically significant correlation, weak and positive, was found between BED D700 cm^3^ and the difference in ALT. A weak positive correlation approaching statistical significance was found between D700 cm^3^ and the difference in ALT, and a weak negative correlation was found between the difference in ALBI and the size of metastases.

A toxicity analysis was also performed according to treatment modality (see Table 7). No statistically significant effect of treatment modality (SBRT vs. brachytherapy) on liver function biochemical parameters was demonstrated.

Late toxicity was analysed in 27 (46%) patients who survived for at least one year without disease progression for nine months. In this group, biochemical parameters indicative of potential liver damage (ALT, AST and total bilirubin) were examined at three-monthly intervals, excluding the period preceding the three-month period prior to radiographic progression of liver metastases. Six patients were assessed once after 6 months; ten patients were assessed twice: after 6 and 9 months. Six patients were assessed three times: after 6, 9 and 12 months. Two patients were assessed four times after 6, 9, 12 and 15 months. One patient was assessed five times after 6, 9, 12, 15 and 18 months. Two patients were assessed six or more times (6, 9, 12, 15, 18, 21, etc.). In this group, grade 1 elevated ALT levels occurred in one measurement in three patients, in two measurements in one patient and in three measurements in two patients. An elevated ALT level (grade 2) was observed in one patient in one measurement. No grade 3 or higher toxicity was observed. An elevated AST level (grade 1) was observed in four patients in one measurement, in two patients in two measurements and in one patient in one measurement. No grade 2 or higher toxicity was observed. An elevated bilirubin level (grade 1) was observed in one measurement in six patients and in two measurements in one patient. No grade 2 or higher toxicity was observed.

## 4. Discussion

This is the first study of its kind to evaluate the role of brachytherapy in reirradiating liver metastases. It includes a selected group of patients who underwent reirradiation after oligoprogression in the previously irradiated area or a different region of the liver. Although both local control and progression-free survival were worse than in the initial treatment, this difference was statistically significant only for local control. The median local control period was 13 months, and the 6-month local control rate was 80%. While this is a worse outcome than that seen in the initial treatment, it is still relatively high given the specificity of reirradiation [16]. Median progression-free survival was 8 months. In cases of oligoprogression or an inability to manage local therapy, the standard approach is to switch to the next line of treatment. The aim of brachytherapy was to maintain the current treatment and thus delay progression. Considering the line of systemic therapy (median–third line), a prolongation of PFS by several months seems to justify local treatment.

The study demonstrated a high disease control rate of 88%, with an average objective response rate of 32%. The most important prognostic factor affecting LC, PFS and OS was the percentage reduction in tumour volume. Studies have confirmed the usefulness of the RECIST score and degree of tumour reduction for prognosis [17,18,19]. Another study on oligopersistent disease in colorectal cancer also identified the degree of tumour reduction after interstitial brachytherapy as a prognostic factor [20]. A study on the reirradiation of primary liver cancer demonstrated that the degree of response to treatment directly impacted OS [10]. Patients who experienced tumour regression after reirradiation had better survival compared to those with stable disease: median overall survival was 30.0 and 4.0 months, respectively (*p* < 0.001). Another significant factor that worsens the prognosis is the number of metastases. Due to re-irradiation, the number of metastases that could be treated was limited to a maximum of four, and most patients had a single metastasis. The study showed that patients with a single metastasis had longer PFS and OS than those with multiple metastases. The study also demonstrated the effect of tumour volume on PFS and OS. In a study by Walter et al. [16], tumour volume was also found to worsen prognosis, although the tumour volumes in that study were significantly smaller than in the present analysis (82 mL vs. 16.4 mL). In a study of colorectal cancer liver metastases, the irradiation volume was also a factor that worsened prognosis [21]. Similar conclusions have been drawn from some studies of SBRT in colorectal cancer [22]. However, other studies have not indicated an effect of metastasis volume on either local control or overall survival [23,24].

Another significant prognostic factor was the dose administered. The study demonstrated longer local control and overall survival after a dose of 25 Gy was administered. Most studies on brachytherapy for liver metastases indicate that high doses are needed to achieve high disease control rates [21,25,26]. Disease control is also better after SBRT with higher doses of radiotherapy [27]. The authors hypothesise that a BED above 100 Gy provides better local control and overall survival than a BED below 100 Gy. Patients in good general condition achieved the greatest benefit, with a statistically significant difference in PFS and OS. Studies on SBRT confirm that this treatment is suitable for patients in good condition with a good prognosis for long-term survival [28]. The study demonstrated the impact of extrahepatic location on OS and PFS. Lung metastases also had an adverse effect on OS. However, the worsening of PFS with lung metastases was not statistically significant, likely due to the small sample size. Data on both brachytherapy and SBRT confirm that extrahepatic location is a negative prognostic factor [21,28,29]. The study also demonstrated the negative impact of treatment at a later stage following multiple progressions. Patients were most often treated during (or after) the third line of systemic therapy, but those treated earlier (lines 1–2) had the best prognosis. This was probably due to the large proportion of patients with colorectal cancer, for which the first two lines of systemic treatment show the greatest activity. These observations are also supported by data on brachytherapy for treating liver metastases from colorectal cancer [30]. Poorer outcomes in future treatments affect how we define the target volume and resistance to chemotherapy [31]. Some authors therefore suggest that local control is worse after prior chemotherapy [31]. However, other authors do not confirm the effect of systemic treatment [32]. As mentioned above, the group of patients included in the analysis was very heterogeneous. The lack of an effect of clinical diagnosis on OS is surprising. This is likely due to the small size of the group and its high level of heterogeneity. The effect of clinical diagnosis on local control was statistically significant. Patients with colorectal cancer have a poorer prognosis than those with breast cancer. However, it is difficult to draw conclusions from comparisons with other cancers, as this group included patients with both indolent tumours (e.g., prostate cancer) and aggressive tumours (e.g., pancreatic cancer). Other data regarding brachytherapy during the first irradiation session confirmed its inferior efficacy in the treatment of colorectal cancer [16]. Studies on the use of SBRT in the treatment of liver metastases have produced conflicting results. This is due to the wide variety of tumours and the small size of patient groups. Some studies indicated statistically significant differences without a negative impact on colorectal cancer [27]. Others indicate a statistically significant effect of diagnosis on overall survival, but without a statistically significant difference in local control. In this case, however, the 3- and 5-year LC in colorectal cancer was the lowest [32]. Others still indicate the effect of diagnosis on local control without affecting overall survival [23]. Similarly to the analysed group of patients, the highest local control results were observed in breast cancer metastases. Numerous phase I–III studies have demonstrated the effect of local therapy in oligometastatic disease [33]. The majority of the data relate to prostate cancer in both the newly diagnosed and oligoprogressive and oligorecurrent stages [34]. The impact of local therapy on prognosis has also been observed in patients with oligometastatic non-small cell lung cancer [34]. However, data regarding the role of local therapy in oligometastatic breast cancer are conflicting [35]. Numerous phase II studies are ongoing in this area [34]. There is also anecdotal data from the treatment of colorectal and renal cancer [34]. An EORTC study confirmed the safety of combining radiotherapy with systemic therapy, although data on this topic are lacking for some therapies (e.g., melanoma) [36]. Different local control depending on the type of cancer results from different tumour biology (e.g., RAS and p53 mutations are considered to increase radioresistance), different sensitivity to radiation expressed by the alpha/beta ratio, possibilities of systemic treatment and the natural history of the cancer [37,38]. The phase III OligoRARE (NCT 04498767) study, which is currently underway, is comparing the addition of radiotherapy to standard systemic therapy in various cancers.

The rate of surgical complications in the study group was relatively low. Grade 1–2 bleeding was observed in 15.5% of patients and grade 3 bleeding in 2%. These results seem comparable to those reported by other authors, although the differences are significant [39]. Grade 3 or higher bleeding occurred in individual patients. Despite repeated radiotherapy, the rate of hepatic complications remained low. Patients were selected so that the BED D2/3 in the unirradiated part of the liver, with an alpha/beta of 3, did not exceed 5 Gy in a single fraction (13.3 Gy). An additional parameter characteristic of stereotactic techniques was also assessed: D700 cm^3^ < 15 Gy administered in three fractions (BED < 40 Gy) [40]. D2/3 was not exceeded in any patient (the maximum BED of the entire treatment was 12.3 Gy) nor was D700 cm^3^ (the maximum BED was 28 Gy). Using such restrictive parameters, with a median time from previous reirradiation of 12 months (range 3–39 months) and a minimum time of nine months for type 1 reirradiation resulted in a low rate of acute hepatic toxicity (grade 1 and 2 toxicity occurred in 37% and 3% of patients, respectively, with no toxicity in higher grades). The study did not demonstrate a worsening of the Child-Pugh scale. The ALBI index worsened in eight patients (13.5%). No signs of RILD were observed. There was no strong correlation between parameters such as dose, irradiated volume, number of metastases or liver dose, and an increase in liver parameters or the ALBI index. The only statistically significant small positive correlation found was between an increase in AST and BED D700 cm^3^, which may indicate a weak correlation between hepatic toxicity and the dose to 700 cm^3^ of the liver. The previous treatment method (SBRT vs. brachytherapy) had no statistically significant effect on the biochemical parameters of the liver. Due to the generally poor prognosis, late liver toxicity only affects a small minority of patients. An analysis of 27 patients who survived for more than a year after brachytherapy, with no progression for nine months, revealed mild hepatic toxicity (grades 1–2). However, these data may be overestimated due to the impact of systemic treatment toxicity. Available studies suggest that reirradiation with SBRT is safe. In the study by Hall et al. [41], no significant hepatic toxicity was observed, and there was no correlation between the ALBI index, the Child-Pugh score, the target volume and the dose. Similarly, hepatic toxicity was relatively common in the study by McDuff [10] but occurred in grades 1–2. A high risk of RILD and associated mortality was associated with baseline liver function [12]. Brachytherapy allows for a reduction in liver dose, as demonstrated by studies comparing brachytherapy with various SBRT techniques [42,43]. Therefore, the hepatic toxicity of brachytherapy is likely to be low [44]. Retrospective analyses of patients with primary liver cancer and liver metastases who were treated with brachytherapy confirm this. RILD occurred in only 0.5% of patients [39]. The risk of RILD is undoubtedly influenced by dose. Analysis by Timmerman et al. [45] indicates that D700 cm^3^ of the liver, administered in one, two, three or five fractions, should not exceed 11.7, 15.1, 17.7 or 21.5 Gy, respectively. Similarly, the QUANTEC study [46] showed that D700 cm^3^ in three to five fraction regimens should not exceed 15 Gy. In our study, D700 cm^3^ administered in one fraction (brachytherapy as reirradiation) was a maximum of 5.9 Gy; with reirradiation (total dose from two to three courses of brachytherapy or three to five SBRT fractions and one or two brachytherapy fractions), the maximum was 10.3 Gy. While these values are significantly lower than the aforementioned tolerance doses, the relationship between liver dose and toxicity requires further investigation, particularly given the heterogeneity of prior treatments.

The study has several limitations. Firstly, it represents a heterogeneous group of patients with various cancers, treatment indications, dosages and types of reirradiation. This undoubtedly impacted the prognosis. Another limitation was the absence of a control group. However, the retrospective nature of the study meant that selection bias could make it difficult to draw definitive conclusions. The patient group included in the analysis is small, meaning that when analysing prognostic factors, some data may not have achieved sufficient statistical power to demonstrate statistically significant differences. Although the short follow-up period does not allow clear conclusions to be drawn regarding late toxicity, the poor prognosis of this patient group means late toxicity appears to be of no clinical significance. Nevertheless, this is the first study of its kind that the authors are aware of. It demonstrates the safety of this procedure and should pave the way for a broader analysis involving a larger, more homogeneous group of patients.

## 5. Conclusions

HDR brachytherapy should be considered as a local treatment for liver metastases in cases of oligometastatic disease, even if SBRT or previous brachytherapy to the liver has been administered. It probably allows for high local control, and the factor that has the greatest impact on local control is the degree of response to treatment, as measured by the percentage of tumour shrinkage. Additionally, the greatest impact on PFS is from a small number of metastases, while the greatest impact on OS is from a small number of metastases and one or two lines of systemic therapy. Furthermore, patients with a good performance status, small-volume metastases confined to the liver, and who receive a dose of 25 Gy are most likely to benefit most from treatment. Local control in colorectal cancer metastases is lower than in other cancers. Brachytherapy treatment is associated with a low rate of serious surgical complications, particularly when a restrictive approach is taken to the dose in the non-irradiated liver parenchyma (BED D2/3 < 13.3 Gy and BED D700 cm^3^ < 40 Gy). This approach is also associated with low early liver toxicity.

## Figures and Tables

**Figure 1 cancers-17-04013-f001:**
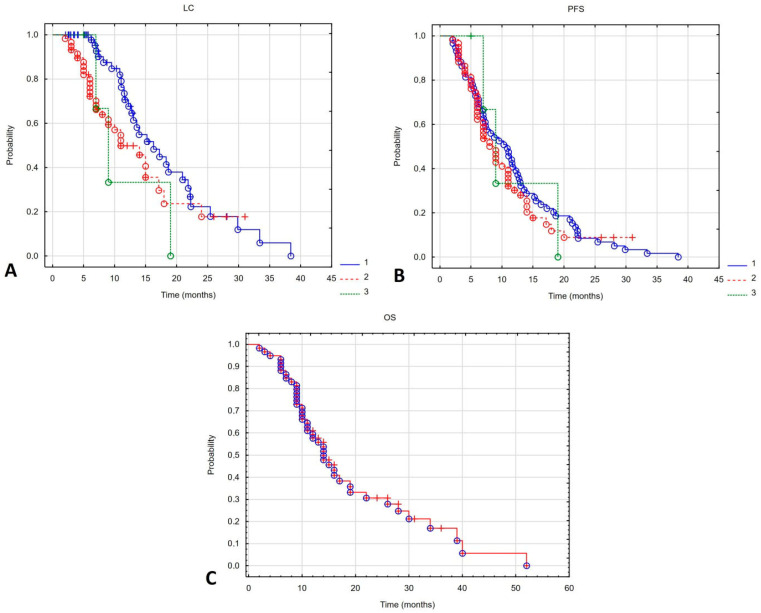
Local control (**A**) and progression-free survival (**B**) depending on the irradiation course (1—first irradiation, 2—second irradiation (first reirradiation), 3—third irradiation (second reirradiation). Kaplan–Meier curve for overall survival of all patients undergoing reirradiation (**C**).

**Figure 2 cancers-17-04013-f002:**
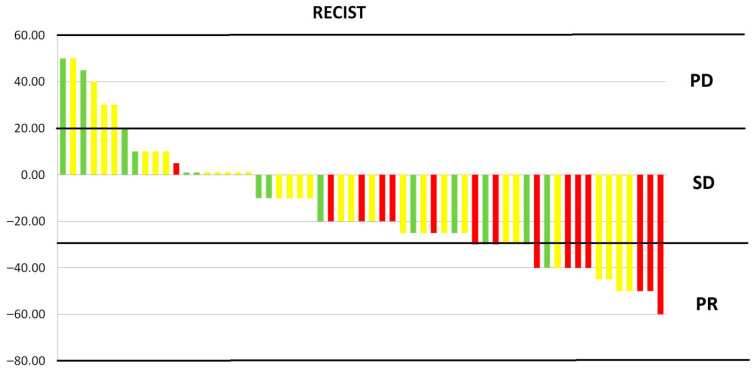
Response to treatment depending on dose. PD, progression disease; SD, stable disease; PR, partial response; red—25 Gy, yellow—20 Gy, green—15 Gy.

**Figure 3 cancers-17-04013-f003:**
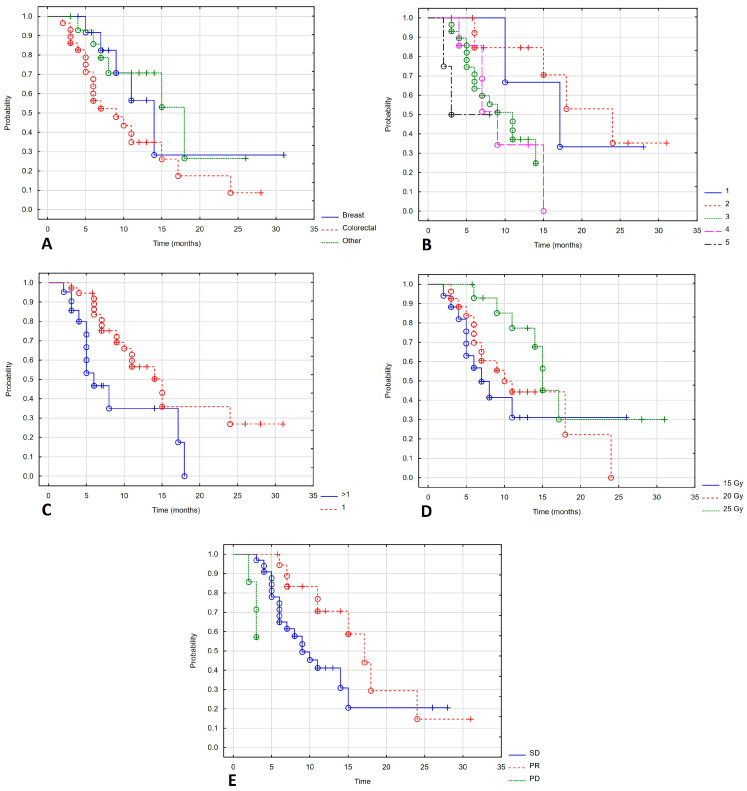
Local control depending on type of cancer (**A**), lines of systemic treatment (**B**), number of metastases (**C**), total dose (**D**), response of treatment (**E**).

**Figure 4 cancers-17-04013-f004:**
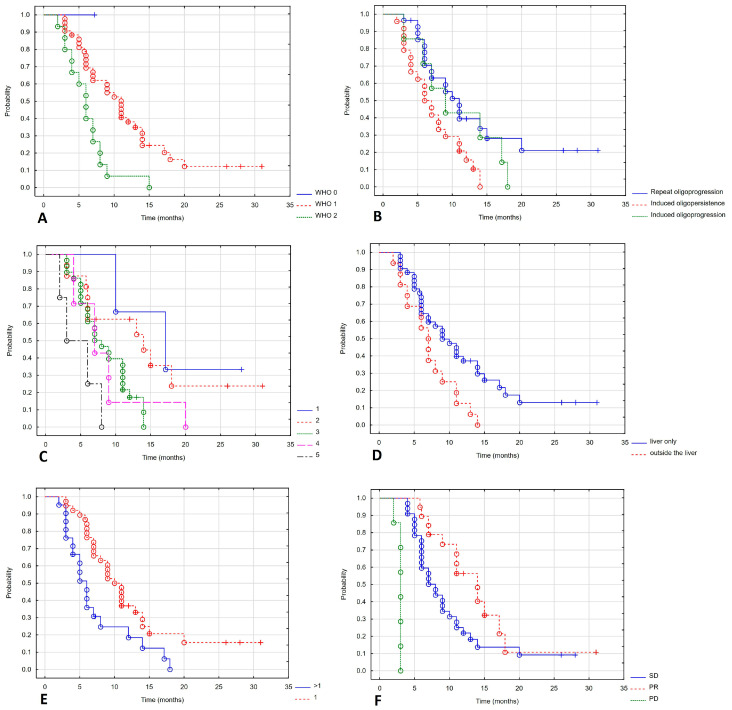
Progression-free survival depending on WHO (**A**), indication of treatment (**B**), line of systemic treatment (**C**), extrahepatic metastases (**D**), number of metastases (**E**), response of treatment (**F**).

**Figure 5 cancers-17-04013-f005:**
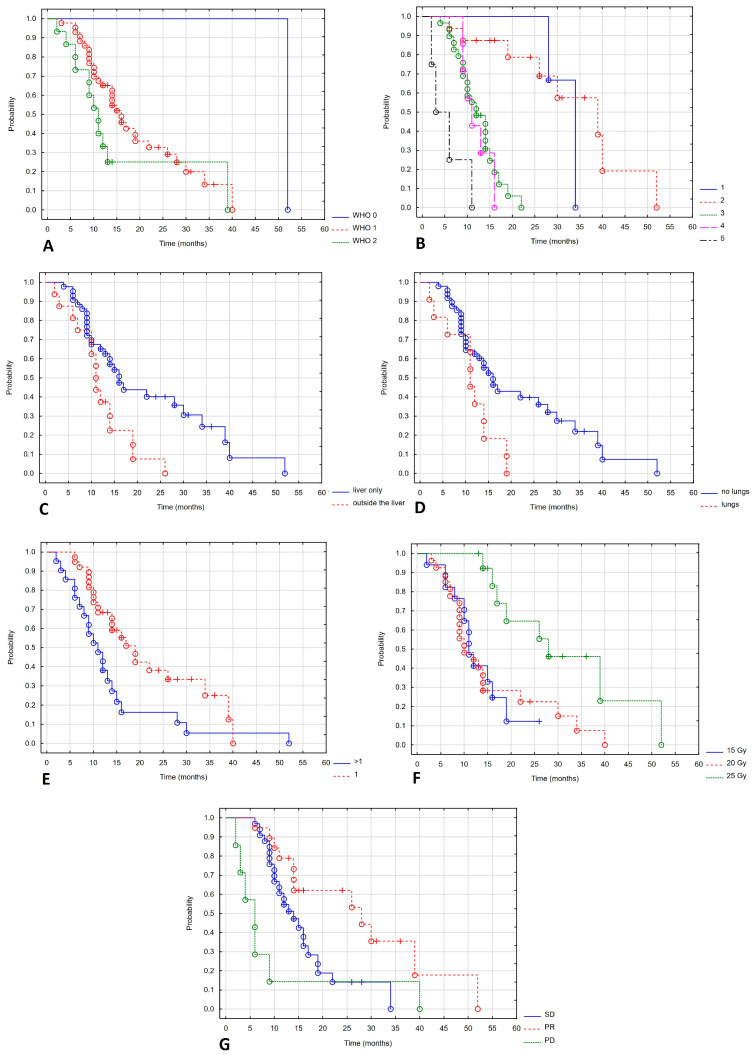
Overall survival depending on WHO (**A**), line of systemic treatment (**B**), extrahepatic and lung metastases (**C**,**D**), number of metastases (**E**), total dose (**F**), response of treatment (**G**).

**Table 1 cancers-17-04013-t001:** Patient characteristics.

Parameter	Number of Patients (Percentage)	Median (Range)
Age		62 (35–82) years
Sex		
Male	31 (52.5%)	
Female	28 (47.5%)	
WHO		
0	1 (2%)	
1	43 (73%)	
2	15 (25%)	
Type of cancer		
Breast	13 (22%)	
Colorectal	30 (51%)	
Gallbladder and extrahepatic cholangiocarcinoma	3 (5%)	
Lung	2 (4%)	
Small intestine cancer	1 (1%)	
Prostate	1 (1%)	
Renal	1 (1%)	
Melanoma	1 (1%)	
Uveal melanoma	1 (1%)	
Endocrine	1 (1%)	
Pancreas	2 (3%)	
Endometrial	1 (1%)	
Unknown	2 (3%)	
Indication		
Repeat oligoprogression	28 (47%)	
Induced oligoprogression	24 (41%)	
Induced oligopersistence	7 (12%)	
Line of treatment		3 (1–5)
1st	3 (5%)	
2nd	16 (27%)	
3rd	29 (50%)	
4th	7 (12%)	
5th	4 (7%)	
Localisation		
Liver only	43 (72%)	
Lung	11 (19%)	
Abdominal	8 (14%)	
Bones	2 (3%)	
Type of reirradiation		
Type 1	29 (49%)	
Type 2	30 (51%)	
Time from previous radiotherapy		12 (3–39) months
Number of tumours		1 (1–4)
Median diameter		4 (1–8)
Volume of all treated tumours		82 (8–513) cm^3^
Fraction dose in previous radiotherapy		20 (6–25) Gy
Number of fractions in previous radiotherapy		1 (1–5)
Total dose in previous radiotherapy		20 (15–50) Gy
BED previous radiotherapy		153 (60–233) Gy
Dose in reirradiation		
15 Gy	17 (29%)	
20 Gy	20 (34%)	
25 Gy	15 (25%)	
BED in reirradiation		100 (60–150) Gy

BED—biologically effective dose.

**Table 2 cancers-17-04013-t002:** Dosimetric analysis of liver.

	In Previous Radiotherapy	In Reirradiation	All
D2/3	1.3 (0.32–3.2) Gy	1.8 (0.4–3.9) Gy	3.15 (0.82–6.1) Gy
D2/3 BED	2.7 (0.6–7.5) Gy	2.88 (0.45–8.97) Gy	5.49 (1.88–12.3) Gy
D700 cm^3^	2.4 (0.4–5.9) Gy	2.3 (0.6–5.1) Gy	4.85 (1.09–10.3) Gy
D700 cm^3^ BED	4.85(2.05–17.5) Gy	4 (0.72–13.77) Gy	10.5 (3.57–28) Gy

D2/3—dose in two-thirds of the non-irradiated liver volume, D700 cm^3^—dose in 700 cm^3^ of the non-irradiated liver volume, BED—biologically effective dose.

**Table 3 cancers-17-04013-t003:** Univariate Cox regression analysis.

Factor	LC			PFS			OS		
	*p*	HR	95%CI	*p*	HR	95%CI	*p*	HR	95% CI
Age of patients	0.451	0.98	0.95–1.02	0.92	1.01	0.97–1.03	0.349	0.98	0.96–1.01
Sex of patients	0.158	0.58	0.27–1.24	0.809	1.07	0.61–1.89	0.879	0.95	0.52–1.74
WHO performance st.	0.118	1.89	0.85–4.25	**0.001**	**2.86**	**1.51–5.39**	**0.012**	**2.23**	**1.19–4.16**
Type of cancer	**0.035**	**2.42**	**0.97–6.05**	0.219	1.54	0.76–3.12	0.276	1.26	0.61–2.60
Indication of treatment	0.529	1.12	0.37–3.35	**0.034**	**1.67**	**0.69–4.04**	0.059	2.01	0.74–5.49
Line of treatment	**0.002**	**1.92**	**1.27–2.91**	**0.001**	**1.68**	**1.23–2.28**	**<0.001**	**2.82**	**1.92–4.12**
Extrahepatic metastases	0.184	0.58	0.26–1.29	**0.018**	**0.47**	**0.25–0.88**	**0.011**	**0.43**	**0.22–0.82**
Lung metastases	0.681	0.81	0.31–2.16	0.091	0.55	0.28–1.09	**0.017**	**0.41**	**0.21–0.85**
Abdominal metastases	0.137	0.49	0.2–1.25	0.196	0.61	0.28–1.32	0.187	0.57	0.25–1.31
Bones metastases	0.989	1.00	1.00–1.00	0.592	0.68	0.16–2.83	0.925	1.09	0.15–8.09
Reirradiation type	0.669	0.856	0.42–1.75	0.373	0.77	0.43–1.37	0.831	0.94	0.51–1.71
Time from initial RT	0.678	1.01	0.97–1.05	0.285	0.98	0.95–1.02	0.134	0.97	0.94–1.01
Number of met.	**0.041**	**1.56**	**1.04–2.34**	**0.002**	**1.66**	**1.2–2.79**	**0.001**	**1.78**	**1.26–2.53**
Average size of tumour	0.504	1.09	0.85–1.4	0.421	1.09	0.88–1.34	0.528	1.08	0.86–1.35
Volume of all tumours	0.191	1.00	1.00–1.01	**0.039**	**1.01**	**1.00–1.01**	**0.046**	**1.00**	**1.00–1.01**
Dose in reirradiation	**0.039**	**2.09**	**0.86–5.05**	0.071	2.02	0.97–4.21	**0.023**	**3.29**	**1.42–7.64**
Tumour reduction	**<0.001**	**1.05**	**1.03–1.07**	**<0.001**	**1.05**	**1.03–1.07**	**<0.001**	**1.02**	**1.01–1.04**

*p*—*p*-value, HR—Hazard Ratio, CI—confidence interval, met.—metastases. Bold font indicates statistically significant values.

**Table 4 cancers-17-04013-t004:** Multivariate Cox regression analysis.

Factor	LC			PFS			OS		
	*p*	HR	95%CI	*p*	HR	95%CI	*p*	HR	95% CI
Line of treatment							**<0.001**	**2.58**	**1.70–3.91**
Number of metastases				**0.015**	**1.53**	**1.09–2.15**	**0.004**	**1.73**	**1.19–2.49**
Tumour reduction	**<0.001**	**1.05**	**1.03–1.07**	**<0.001**	**1.05**	**1.03–1.07**	**<0.001**	**1.02**	**1.01–1.03**

*p*—*p*-value, HR—Hazard Ratio, CI—confidence interval. Bold font indicates statistically significant values.

**Table 5 cancers-17-04013-t005:** Frequency of acute toxicity according to RECIST grade.

Parameter	CTCA 1	CTCA 2	CTCA3
Bleeding	8 (13.5%)	2 (3%)	1 (2%)
Oedema	1 (2%)	0	0
Pain	14 (25%)	0	0
Stomatitis	0	1 (2%)	0
Skin reaction	2 (1%)	0	0
Liver dysfunction (total)	22 (37%)	2 (3%)	0
increased bilirubin levels	6 (10%)	0	0
increased ALT levels	19 (32%)	2 (3%)	0
increased AST levels	15 (25%)	0	0

AST—aspartate aminotransferase, ALT—alanine aminotransferase.

**Table 6 cancers-17-04013-t006:** Correlation between dosimetric parameters of the treatment plan and biochemical parameters of liver function after brachytherapy and difference in values before and after brachytherapy. Each position shows the Pearson correlation coefficient value and *p*-value.

	Time from Previous Radiotherapy	Dose	Number of Metastases	Mean Diameter	Volume	D2/3	D2/3 BED	D700 cm^3^	D700 cm^3^ BED
ALBI post treatment	−0.05 *p* = 0.669	−0.037 *p* = 0.781	−0.212 *p* = 0.106	0.164 *p* = 0.214	0.014 *p* = 0.919	0.056 *p* = 0.673	0.043 *p* = 0.747	0.147 *p* = 0.266	0.093 *p* = 0.483
ALBI difference	0.121 *p* = 0.360	0.113 *p* = 0.394	0.138 *p* = 0.298	−0.245 *p* = 0.062	−0.155 *p* = 0.241	−0.206 *p* = 0.118	−0.1928 *p* = 0.144	−0.100 *p* = 0.450	−0.0940 *p* = 0.479
Bilirubin post treatment	−0.042 *p* = 0.752	−0.060 *p* = 0.654	0.030 *p* = 0.819	0.011 *p* = 0.934	−0.080 *p* = 0.546	0.040 *p* = 0.765	0.043 *p* = 0.747	0.040 *p* = 0.765	−0.012 *p* = 0.882
Bilirubin difference	0.169 *p* = 0.202	0.189 *p* = 0.151	−0.068 *p* = 0.611	0.148 *p* = 0.264	0.056 *p* = 0.672	0.042 *p* = 0.753	0.167 *p* = 0.207	0.042 *p* = 0.753	0.062 *p* = 0.639
ALT post treatment	−0.210 *p* = 0.110	−0.166 *p* = 0.208	−0.028 *p* = 0.836	0.109 *p* = 0.410	0.053 *p* = 0.690	0.212 *p* = 0.107	0.154 *p* = 0.245	0.212 *p* = 0.107	0.231 *p* = 0.078
ALT difference	−0.167 *p* = 0.208	−0.142 *p* = 0.282	0.080 *p* = 0.547	0.026 *p* = 0.848	−0.003 *p* = 0.982	0.247 *p* = 0.059	0.153 *p* = 0.246	0.247 *p* = 0.059	**0.267** ** *p* ** **= 0.041**
AST post treatment	−0.187 *p* = 0.157	−0.233 *p* = 0.076	0.066 *p* = 0.619	−0.058 *p* = 0.661	−0.190 *p* = 0.150	−0.068 *p* = 0.611	0.031 *p* = 0.814	−0.068 *p* = 0.611	−0.057 *p* = 0.666
AST difference	−0.085 *p* = 0.521	−0.182 *p* = 0.168	0.095 *p* = 0.477	−0.236 *p* = 0.072	−0.019 *p* = 0.888	−0.165 *p* = 0.212	−0.189 *p* = 0.150	−0.165 *p* = 0.212	−0.200 *p* = 0.128
Albumin post treatment	0.055 *p* = 0.681	0.016 *p* = 0.904	0.212 *p* = 0.107	0.211 *p* = 0.109	−0.045 *p* = 0.734	−0.049 *p* = 0.713	0.058 *p* = 0.664	−0.146 *p* = 0.270	−0.109 *p* = 0.410
Albumin difference	−0.181 *p* = 0.169	−0.172 *p* = 0.192	−0.115 *p* = 0.387	−0.169 *p* = 0.201	0.144 *p* = 0.277	0.193 *p* = 0.144	0.164 *p* = 0.215	−0.100 *p* = 0.450	0.085 *p* = 0.524

D2/3—dose in two-thirds of the non-irradiated liver volume, D700 cm^3^—dose in 700 cm^3^ of the non-irradiated liver volume, BED—biologically effective dose, AST—aspartate aminotransferase, ALT—alanine aminotransferase, Bold font indicates statistically significant values.

**Table 7 cancers-17-04013-t007:** Toxicity analysis by treatment method.

	SBRT Median (Range)	Brachytherapy Median (Range)	U-Mann–Whitney Test Value	*p*-Value
ALBI post treatment	−2.735 (−3.296–1.799)	−2.586 (−3.211–1.809)	1.353	0.176
ALBI difference	0 (−0.991–0.352)	−0.036 (−1.588–0.68)	−0.365	0.714
Bilirubin post treat.	0.9 (0.5–2) mg/dL	1 (0.4–2.1) mg/dL	0.457	0.647
Bilirubin difference	0 (−0.8–1.3) mg/dL	0 (−0.4–1.1) mg/dL	−0.667	0.504
ALT post treatment	43 (24–65) IU/L	43 (13–175) IU/L	0.109	0.913
ALT difference	11 (−9–28) IU/L	10.5 (−23–133) IU/L dL	0.932	0.351
AST post treatment	43 (18–55) IU/L	36 (21–89) IU/L	−0.338	0.735
AST difference	8 (−2–28) IU/L	9.5 (−8–48) IU/L	0.512	0.609
Albumin post treat.	4.1 (3–4.8) g/dL	4 (3.1–4.7) g/dL	−1.189	0.235
Albumin difference	0.2 (−0.6–1.4) g/dL	0.1 (−0.8–1.6) g/dL	0.270	0.787

AST—aspartate aminotransferase, ALT—alanine aminotransferase, SBRT—stereotactic body radiation therapy, treat.—treatment.

## Data Availability

The raw data supporting the conclusions of this article will be made available by the authors on request.

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
