# Peer review of "The Role of High-Dose-Rate Brachytherapy (Interventional Radiotherapy) in the Reirradiation of Liver Metastases"

_cancers, 2025, doi:10.3390/cancers17244013_

Round 1
Reviewer 1 Report
Comments and Suggestions for Authors
The topic fills an important gap, as data on liver reirradiation with HDR brachytherapy are scarce. The manuscript is thorough, but the density of information makes some sections pretty hard to follow. Overall structure is standard and appropriate, though several areas could be tightened or clarified.
Major concerns that require revision:
1. Cohort heterogeneity - The patient mix (tumor types, indications, treatment history) is very broad. This needs stronger discussion about how it may confound survival and LC findings.
2. Terminology clarity - The manuscript uses several oligometastatic subtypes (repeat oligoprogression, induced oligopersistence). These should be briefly redefined in the Introduction, not only Methods.
3. Strenghtening the Introduction - The majority of liver mets come from colorectal. Please expand this topic in the Introduction and include explanation of the development of M1hep after insufficient colorectal cancer treatment. A great example are neoadjuvant treatment interruptions presented in this article https://doi.org/10.3390/jpm14030266
Consider discussing it in the Introduction.
4. Outcome interpretation - LC, PFS, and OS numbers are sometimes presented multiple times with slight variation; consolidating them would improve readability.
5. Tables - some tables (1, 3, 5, 6) have visible flaws. For example table 1 uses bulleted and centered phrases in column 1 and are difficult to follow. Table 3, column 1 is too narrow and difficult to follow, and it also lacks black lines between the rows for better readability. Table 5 is hard to follow especially in the last 3 rows. Table 6 is difficult to follow.
6. Please be careful with the English language used. Example of errors: "bledding" instead of "bleeding" in table 5.
Minor comments:
- Several sentences are long and could be shortened for clarity, particularly in the Introduction and Discussion.
-A few numerical results are repeated across sections; streamlining would help the flow.
- Typographical and formatting issues appear throughout, including spacing and inconsistent use of abbreviations.
The final section is reasonable but could more clearly distinguish between findings that are strongly supported by the data and those that are hypothesis-generating due to the small sample size and heterogeneity.
Overall this study has potential for publishing after major revisions.
Comments on the Quality of English LanguagePlease be careful with the English language used. Example of errors: "bledding" instead of "bleeding" in table 5.
Author Response
Comments 1:
Cohort heterogeneity - The patient mix (tumor types, indications, treatment history) is very broad. This needs stronger discussion about how it may confound survival and LC findings.
Response 1
We agree with your comment. We have added a comprehensive section on how tumour type and clinical situation impact treatment outcomes with radiotherapy in oligometastatic disease.
Comments 2
Terminology clarity - The manuscript uses several oligometastatic subtypes (repeat oligoprogression, induced oligopersistence). These should be briefly redefined in the Introduction, not only Methods.
Response 2
Thank you for bringing this to our attention. We have added brief definitions of 'repeat oligoprogression', 'induced oligoprogression' and 'oligopersistence disease' to the introduction.
Comments 3
Strenghtening the Introduction - The majority of liver mets come from colorectal. Please expand this topic in the Introduction and include explanation of the development of M1hep after insufficient colorectal cancer treatment. A great example are neoadjuvant treatment interruptions presented in this article https://doi.org/10.3390/jpm14030266
Response 3
Thank you for bringing this to our attention. We have added a section to the introduction that discusses the impact of suboptimal induction therapy on the occurrence of liver metastases from colorectal cancer.
Comments 4
Outcome interpretation - LC, PFS, and OS numbers are sometimes presented multiple times with slight variation; consolidating them would improve readability.
Response 4
We have consolidated the first subsection of the results to improve their readability. We have shortened and divided this section into smaller parts to make it clearer and easier to understand.
Comments 5
Tables - some tables (1, 3, 5, 6) have visible flaws. For example table 1 uses bulleted and centered phrases in column 1 and are difficult to follow. Table 3, column 1 is too narrow and difficult to follow, and it also lacks black lines between the rows for better readability. Table 5 is hard to follow especially in the last 3 rows. Table 6 is difficult to follow.
Response 5
We agree with You
The errors in the tables are partly due to the editorial team reformatting the tables before submitting them for review. These errors were not present in our original version. Similarly, the lack of black lines between rows is due to the table layout proposed by the editorial team. We have corrected all errors, but we cannot guarantee that they will not be changed again before submitting them for further review.
Table 6 has been reformatted to a horizontal page. A detailed description has been added to the table title.
Comments 6
Please be careful with the English language used. Example of errors: "bledding" instead of "bleeding" in table 5.
Response 6
Thank you for bringing this to our attention. We've checked the text for grammatical errors.
Comments 7
Several sentences are long and could be shortened for clarity, particularly in the Introduction and Discussion
Response 7
Some long sentences of text from the Introduction and Discussion have been removed, shortened, or moved.
Comment 8
A few numerical results are repeated across sections; streamlining would help the flow. Typographical and formatting issues appear throughout, including spacing and inconsistent use of abbreviations.
Response 8
Some duplicate results have been removed to make the text clearer. Some sections in the results section have been shortened to make the text more readable. The text has been improved with regard to formatting and the consistency of abbreviations used.
Comment 9
The final section is reasonable but could more clearly distinguish between findings that are strongly supported by the data and those that are hypothesis-generating due to the small sample size and heterogeneity.
Response 9
The conclusions have been revised to make it easier to distinguish between findings that are strongly supported by data and those that generate hypotheses.

Reviewer 2 Report
Comments and Suggestions for Authors
-
eterogeneous Patient Group: The study includes patients with a wide variety of cancers and treatment backgrounds, which makes it difficult to draw definitive conclusions. The small, heterogeneous sample size reduces the statistical power of the analysis, especially when attempting to assess prognostic factors across different cancer types.
-
Limited Statistical Significance in Some Results: While certain factors like tumor shrinkage, dose, and performance status significantly influenced outcomes, many other important variables (such as type of reirradiation or extrahepatic metastases) showed weak correlations with overall survival (OS) and progression-free survival (PFS). Some analyses did not achieve statistical significance, which suggests that the sample size may not be sufficient for meaningful conclusions on these factors.
-
Lack of Long-term Follow-up Data: The median follow-up of 13 months provides useful data, but it may not be sufficient for assessing the long-term efficacy and safety of HDR brachytherapy for liver metastases. Longer-term follow-up is needed to evaluate potential late toxicities, recurrence rates, and survival trends beyond the short-term period.
-
Absence of a Control Group: The study lacks a control group of patients who received alternative treatments. Without a comparative analysis, it’s difficult to conclude whether HDR brachytherapy truly provides better outcomes than other forms of radiotherapy or systemic therapies.
-
Limited Discussion on Toxicity and Complications: The report mentions low toxicity rates and complications, but a deeper analysis of the potential risks, long-term liver function effects, and comparisons to other radiation therapies (like SBRT) is lacking. For instance, the relationship between liver dose and toxicity needs further exploration, especially given the heterogeneity of prior treatments.
-
No Clear Stratification Based on Cancer Type: Although the study mentions differences in treatment response based on cancer type (e.g., colorectal cancer vs. breast cancer), it does not provide a thorough breakdown of how these differences affect the treatment outcomes in detail. This should be expanded upon, as cancer type could significantly alter treatment efficacy.
Author Response
|
Comments 1: Heterogeneous Patient Group: The study includes patients with a wide variety of cancers and treatment backgrounds, which makes it difficult to draw definitive conclusions. The small, heterogeneous sample size reduces the statistical power of the analysis, especially when attempting to assess prognostic factors across different cancer types.
|
|
Response 1: We agree with this comment. The cohort is heterogeneous and not very large. This is emphasised in the study limitations. The treatment of liver metastases with reirradiation, regardless of the technique used, is poorly understood. To date, the largest published study includes only 49 patients, of whom 23 metastases had various diagnoses (DOI: 10.1016/j.prro.2018.04.012). The remaining analyses are case reports only. Various EBRT techniques were employed in these analyses. This is the first time that brachytherapy treatment has been described. This analysis aims to provide a preliminary assessment of the efficacy and toxicity of brachytherapy, and we believe it should be the starting point for further analysis in larger, more homogeneous patient groups. In the introduction, we mentioned that these are preliminary treatment results.
Comments 2 Limited Statistical Significance in Some Results: While certain factors like tumor shrinkage, dose, and performance status significantly influenced outcomes, many other important variables (such as type of reirradiation or extrahepatic metastases) showed weak correlations with overall survival (OS) and progression-free survival (PFS). Some analyses did not achieve statistical significance, which suggests that the sample size may not be sufficient for meaningful conclusions on these factors.
Response 2 We agree that the small sample size may not be sufficient to achieve statistical significance for many important prognostic factors. This fact is emphasised in the study limitations. Many brachytherapy studies are based on small patient groups because the indications for brachytherapy are quite narrow. Randomised phase II trials investigating the role of SBRT in liver metastases also include a similar number of patients (DOI: 10.1186/s13014-018-1185-9). Some phase II SBRT analyses (e.g. SABR-COMET) are only slightly larger and are significantly more heterogeneous in terms of diagnosis and metastasis location. Meanwhile, SBRT is a well-established and widely used treatment for liver metastases worldwide. As previously mentioned, there are no large-scale studies examining the role of reirradiation in liver metastases. This is primarily due to the expected toxicity of the treatment. Brachytherapy has an undoubted advantage in protecting organs at risk (OARs), as demonstrated in numerous studies cited in the manuscript (e.g. https://doi.org/10.1016/j.phro.2025.100811). Unfortunately, the lack of large-scale analyses, the need for appropriate equipment (specialised CT or MR imaging capable of being applied during imaging) and a properly trained team makes recruiting a larger group of patients very difficult. Our experience spans 12 years and includes over 600 patients who have been treated with brachytherapy to date, as well as hundreds of patients who have been treated with SBRT. Therefore, a group of 59 patients still seems relatively large. We plan to recruit a larger group of patients in the future to ensure the results are statistically significant.
Comments 3 Lack of Long-term Follow-up Data: The median follow-up of 13 months provides useful data, but it may not be sufficient for assessing the long-term efficacy and safety of HDR brachytherapy for liver metastases. Longer-term follow-up is needed to evaluate potential late toxicities, recurrence rates, and survival trends beyond the short-term period.
Response 3 We agree that the median follow-up period appears short, but this is due to the poor prognosis of the analysed patient group. Patients who underwent reirradiation had received multiple lines of systemic therapy for metastatic cancer. The median overall survival (OS) in the study group was 14 months, and progression-free survival (PFS) was 8 months. Only 14 patients (23%) survived until the end of the follow-up period. Furthermore, only 11 patients (18%) showed no signs of progression by the end of the follow-up period. Therefore, most patients did not survive long enough to experience potential late toxicity.
We have added a section analysing late toxicity in patients who survived for at least one year after reirradiation. In this group, biochemical parameters of potential liver damage (ALT, AST, total bilirubin) were examined at three-monthly intervals, excluding the period preceding the three-month period prior to radiographic progression of liver metastases. However, these data may be overestimated due to the toxic effects of systemic treatment.
Comments 4 Absence of a Control Group: The study lacks a control group of patients who received alternative treatments. Without a comparative analysis, it’s difficult to conclude whether HDR brachytherapy truly provides better outcomes than other forms of radiotherapy or systemic therapies.
Response 4 Thank you for pointing out this limitation of the study. The lack of a control group has been added to the list of study limitations in the discussion section. While we agree that the lack of a control group makes it difficult to draw conclusions regarding the superiority of brachytherapy over other radiotherapy methods or systemic treatment, in the case of reirradiation there is no standard treatment involving another radiotherapy method. The available data only concern SBRT; however, as mentioned above, it is difficult to consider SBRT a standard method in this case. This is due to the lack of large randomised clinical trials of SBRT for the re-irradiation of liver metastases. The use of SBRT in this group of patients may also be associated with higher toxicity due to the size of the irradiated metastases. SBRT is generally recommended for metastases no larger than 4 cm, and this applies to the initial irradiation too. In our analysis, the median size was 4 cm (range 1–8 cm), and the volume was 82 cm³ (range 8–513 cm³), which is significantly larger than the size permitted for SBRT (DOI: 10.1186/s13014-018-0969-2). Adding a control group of systemic therapy alone would have introduced patient selection bias, since in cases of oligoprogression or an inability to manage local therapy, the standard approach is to switch to the next line of treatment. The aim of brachytherapy was to maintain the current treatment and thus delay progression. Therefore, it seems that using a control group without local treatment or reirradiation with SRBT, with the option of brachytherapy, would offer patients a worse treatment option
Comments 5 Limited Discussion on Toxicity and Complications: The report mentions low toxicity rates and complications, but a deeper analysis of the potential risks, long-term liver function effects, and comparisons to other radiation therapies (like SBRT) is lacking. For instance, the relationship between liver dose and toxicity needs further exploration, especially given the heterogeneity of prior treatments.
Response 5 Thank you for highlighting this aspect of the study. We have added an analysis of how the irradiation method affects biochemical liver function parameters. This topic has also been expanded upon in the discussion section. We have added a section on the effect of dose on liver toxicity.
Comments 6 No Clear Stratification Based on Cancer Type: Although the study mentions differences in treatment response based on cancer type (e.g., colorectal cancer vs. breast cancer), it does not provide a thorough breakdown of how these differences affect the treatment outcomes in detail. This should be expanded upon, as cancer type could significantly alter treatment efficacy.
Response 6 We agree with this suggestion. The role of radiotherapy in oligometastatic disease has been proven in some cancers, such as prostate and lung cancer. In others, such as breast cancer, the data are inconclusive. We have added a section to the discussion on the impact of cancer type on treatment outcomes. The need for further research on this topic has also been emphasised.
The English could be improved to more clearly express the research.
Response: The English language has been checked and any errors have been corrected.
|

Reviewer 3 Report
Comments and Suggestions for Authors
This study analyzes treatment outcomes, identifies prognostic factors, and assesses the toxicity of HDR brachytherapy reirradiation for liver metastases in patients with oligometastatic disease. The work has a certain degree of novelty. Below are my specific comments:
- The title should avoid abbreviations whenever possible. Please consider using full terms to improve clarity and academic rigor.
- At line 213, Figure 1 is mentioned without any accompanying description. Similarly, at line 223, only “Figure 2” appears without explanation. Please provide appropriate descriptions in the text.
- At line 235, the expression “Tables 3 and 4” appears twice and should be corrected.
- After the paragraph ending at line 249, the sudden appearance of “(Fig.2.)” is unclear and seems to be an unintended or misplaced citation. Please revise.
- Figures 3–5 are not adequately described in the main text. Additional explanations and interpretations should be provided.
- The subheadings within the Materials and Methods section contain numbering errors.
- There is a typographical error in the dose description at line 245: “for doses of 25 Gy, 20 Gy and 25 Gy.” Please verify and correct.
- In Figures 4 and 5, the sizes of the subfigures are inconsistent. Please standardize the dimensions and align the subfigures to improve readability and presentation quality.
- The Discussion section would benefit from structural reorganization: Lines 347–358 primarily discuss the role of SBRT in reirradiation, which represents background information and is more suitable for the Introduction. Lines 361–379 also summarize previous studies and should be moved to the Introduction. Lines 383–402 again present background information regarding SBRT and should likewise be relocated to the Introduction. Lines 403–427 do not directly discuss the findings of the present study and fit better in the Introduction. Overall, the Discussion section should focus more on interpreting the results of this study and comparing them with relevant literature on HDR reirradiation.
Author Response
|
Comments 1 |
|
The title should avoid abbreviations whenever possible. Please consider using full terms to improve clarity and academic rigor. Response 1 Thank you for this suggestion. We have changed the manuscript title to avoid abbreviations.
Comments 2 At line 213, Figure 1 is mentioned without any accompanying description. Similarly, at line 223, only “Figure 2” appears without explanation. Please provide appropriate descriptions in the text.
Response 2 Thank you for pointing this out. A description of the figure has been added to the text.
Comments 3 At line 235, the expression “Tables 3 and 4” appears twice and should be corrected.
Response 3 Thank you for bringing this to our attention. The duplicate entry has been removed.
Comments 4 After the paragraph ending at line 249, the sudden appearance of “(Fig.2.)” is unclear and seems to be an unintended or misplaced citation. Please revise.
Response 4 Thank you for bringing this to our attention. This text was written by mistake. It has been deleted.
Comments 5 Figures 3–5 are not adequately described in the main text. Additional explanations and interpretations should be provided.
Response 5 We have added a detailed description of figures 3-5 in the manuscript text.
Comments 6 The subheadings within the Materials and Methods section contain numbering errors.
Response 6 Thank you for pointing this out. We have corrected the numbering of the subheadings.
Comments 7 There is a typographical error in the dose description at line 245: “for doses of 25 Gy, 20 Gy and 25 Gy.” Please verify and correct.
Response 7 Thank you for bringing this to our attention. We've corrected this section of text.
Comments 8 In Figures 4 and 5, the sizes of the subfigures are inconsistent. Please standardize the dimensions and align the subfigures to improve readability and presentation quality.
Response 8 We have corrected figures 3, 4 and 5 to have the same dimensions
Comments 9 The Discussion section would benefit from structural reorganization: Lines 347–358 primarily discuss the role of SBRT in reirradiation, which represents background information and is more suitable for the Introduction. Lines 361–379 also summarize previous studies and should be moved to the Introduction. Lines 383–402 again present background information regarding SBRT and should likewise be relocated to the Introduction. Lines 403–427 do not directly discuss the findings of the present study and fit better in the Introduction. Overall, the Discussion section should focus more on interpreting the results of this study and comparing them with relevant literature on HDR reirradiation.
Response 9 Thank you for your suggestion. The section on reirradiation with SBRT has been revised and relocated to the introduction. The remaining sections have been shortened to focus more closely on the study's topic. |

Round 2
Reviewer 1 Report
Comments and Suggestions for Authors
All of my concerns were appropriately handled.
The article is worth publishing.
Reviewer 2 Report
Comments and Suggestions for Authors
Accept